# Evaluation of Four Satellite Precipitation Products over Mainland China Using Spatial Correlation Analysis

**Yu Li** [1,2], **Bo Pang** [1,2,*], **Ziqi Zheng** [1,2], **Haoming Chen** [1,2], **Dingzhi Peng** [1,2], **Zhongfan Zhu** [1,2] **and Depeng Zuo** [1,2]

1   College of Water Sciences, Beijing Normal University, Beijing 100875, China
2   Beijing Key Laboratory of Urban Hydrological Cycle and Sponge City Technology, Beijing 100875, China
*   Correspondence: pb@bnu.edu.cn; Tel.: +86-135-2168-6445

**Abstract:** The accuracy and reliability of satellite precipitation products (SPPs) are important for their applications. In this study, four recently presented SPPs, namely, GSMaP_Gauge, GSMaP_NRT, IMERG, and MSWEP, were evaluated against daily observations from 2344 gauges of mainland China from 2001 to 2018. Bivariate Moran's I (BMI), a method that has demonstrated high applicability in characterizing spatial correlation and dependence, was first used in research to assess their spatial correlations with gauge observations. Results from four conventional indices indicate that MSWEP exhibited the best performance, with a correlation coefficient of 0.78, an absolute deviation of 1.6, a relative bias of −5%, and a root mean square error of 5. Six precipitation indices were selected to further evaluate the spatial correlation between the SPPs and gauge observations. MSWEP demonstrated the best spatial correlation in annual total precipitation, annual precipitation days, continuous wet days, continuous dry days, and very wet day precipitation with global BMI of 0.95, 0.78, 0.78, 0.78, and 0.87, respectively. Meanwhile, IMERG showed superiority in terms of maximum daily precipitation with a global BMI value of 0.91. IMERG also exhibited superior performance in quantifying the annual count days that experience precipitation events exceeding 25 mm and 50 mm, with a global BMI of 0.96, 0.92. In four sub-regions, these products exhibited significant regional characteristics. MSWEP demonstrated the highest spatial correlation with gauge observations in terms of total and persistent indices in the four sub-regions, while IMERG had the highest global BMI for extreme indices. In general, global BMI can quantitatively compare the spatial correlation between SPPs and gauge observations. The Local Indicator of Spatial Association (LISA) cluster map provides clear visual representation of areas that are significantly overestimated or underestimated. These advantages make BMI a suitable method for SPPs assessment.

**Keywords:** satellite precipitation products; Bivariate Moran's I; LISA cluster map; GSMAP; IMERG; MSWEP

## 1. Introduction

The territory of China is characterized by a wide range of geographical conditions and climatic zones, which result in a complex and varied distribution of precipitation across the region. The eastern areas of China are influenced by a monsoonal climate, while the northwestern regions have a temperate continental climate, and the Tibet Plateau have an alpine climate [1,2]. These complexities in precipitation patterns influence the frequency of drought and flood events significantly, which pose a threat to human safety and well-being, food security, and the stability of the ecosystem [3]. Thus, reliable and high-resolution quantitative precipitation estimation is of the utmost importance for effective risk management and climate change adaption.

Precipitation observations can be achieved with three main methods: gauge observations, weather radars, and satellite sensors [4]. As a traditional precipitation observation method, the advantages of the gauge observations include accurate point data and long

historical data. However, the uneven distribution of rain gauges in China would lead to large errors in the spatial interpolation method for obtaining spatially continuous data in areas with a low rainfall station density and complex terrain [5]. Weather radars can provide three-dimensional precipitation observations with temporal and spatial resolutions of minutes and kilometers, respectively [6]. However, the spatial coverage of radars is limited, and signal occlusion by complex terrain, signal attenuation, and the inaccuracies of the reflectivity-precipitation rate (Z-R) relationship leads to uncertainty in precipitation estimates [7]. Currently, satellite sensors can provide a uniform precipitation measurement globally, compensating for the limitations of rain gauge observations with its wide spatial coverage and continuous precipitation measurement [8].

At present, the two landmark satellites for precipitation measurement are the Tropical Rainfall Measuring Mission (TRMM) [9] satellite and the Global Precipitation Measurement (GPM) satellite [10,11]. GPM is a follow-up plan of TRMM, which can quantify the microscopic physical properties of precipitation particles and detect light intensity precipitation and snow more accurately. With the advantages of satellite sensors, a variety of high resolution satellite precipitation products (SPPs) have been developed [12–17], including Integrated Multi-Satellite Retrievals for GPM (IMERG), Global Precipitation Satellite Mapping products (GSMaP), Climate Prediction Center Morphing Technique (CMORPH) [16], Multi-Source Weighted-Ensemble Precipitation (MSWEP) [17], etc., which provide researchers with a variety of options.

The evaluation of SPPs is essential for its applications, which include both global assessments [18–20] and regional assessments [21–25] covering different time scales [26–30]. Statistical indices such as the absolute deviation (AD), relative deviation (RB), root mean square error (RMSE), and correlation coefficient (Corr) are most common indices in these studies [31–34]. In recent years, extreme indices, which are important for flood prevention and drought management, have received significant attention and have been studied by many researchers [35–41]. However, most of these indices are grid-based and are calculated based on the differences between gauge observations and SPP grids with the same geographical locations. These indices can evaluate the performance of SPPs in a single grid and their statistical values, such as the mean value, median value, and variance, and can also provide an overview of the total performance in a region. Nevertheless, evaluating the performance of SPPs with respect to spatial distribution remains a challenging task. Most evaluations rely on the figure description without explicit indices. Recently, a map comparison technique, the Structural Similarity Index (SSI) [42], was used to identify differences between two spatial distribution maps of precipitation. However, as a map-based technique, SSI requires gauge observations to be transferred to grid maps through interpolation, which restrict its application.

The Bivariate Moran's I (BMI) [43] is a widely adopted spatial correlation index that represents a spatial weighting of Pearson's correlation coefficient. The index possesses high applicability and effectiveness in capturing the spatial correlation and interdependence between two elements. The BMI is a well-established indicator of spatial correlation and has been extensively utilized in evaluating the spatial relationships between different geographical factors [44–46]. Furthermore, the Local Indicator of Spatial Association (LISA) cluster maps can be utilized to determine the regional distribution of local correlation types and their statistical significance. Therefore, we believe that the BMI is very suitable for evaluating the spatial correlation between SPPs and gauge observations.

The objective of this study is to evaluate the performance of four recent SPPs over mainland China against daily observations from 2344 gauges between 2001 to 2018. The four SPPs include the current GPM products, GSMaP-Gauge, GSMaP-NRT, IMERG, and the widely used precipitation fusion product, MSWEP. A novel spatial correlation index, BMI, is adopted for the first time to evaluate the SPPs with respect to spatial distribution. The specific objectives include: (1) to assess the performance of the four SPPs in terms of absolute deviation, relative bias, root mean square error, and correlation coefficient against the gauge observations; (2) to use the BMI to identify the spatial correlation between six

precipitation indices obtained from SPPs and those from gauge observations in mainland China and its sub-regions, and to use the LISA cluster map to analyze the local clustering characteristics of their differences; and (3) to compare the advantages of the BMI in spatial assessment with the universal assessment method. The study provides a comprehensive evaluation of the overall spatial correlation and local clustering characteristics of SPPs over mainland China and expands the spatial evaluation methods.

## 2. Materials and Methods

### 2.1. Study Area

Due to the large differences in natural geographical conditions and contrasts in climate, precipitation in mainland China is highly variable and characterized by complex spatial and temporal distributions [47]. Therefore, we divide mainland China into subregions to capture the regional characteristics of precipitation [48]. According to the altitude, annual precipitation distribution, and the existence of mountains, it can be divided into three regions: the eastern monsoon region, the northwest region (NWC), and the Tibet Plateau region (TP). The eastern monsoon region is further divided into the northern region (NC) and the southern region (SC) according to the latitude difference, separated by the Qinling Mountains Huaihe River (Figure 1).

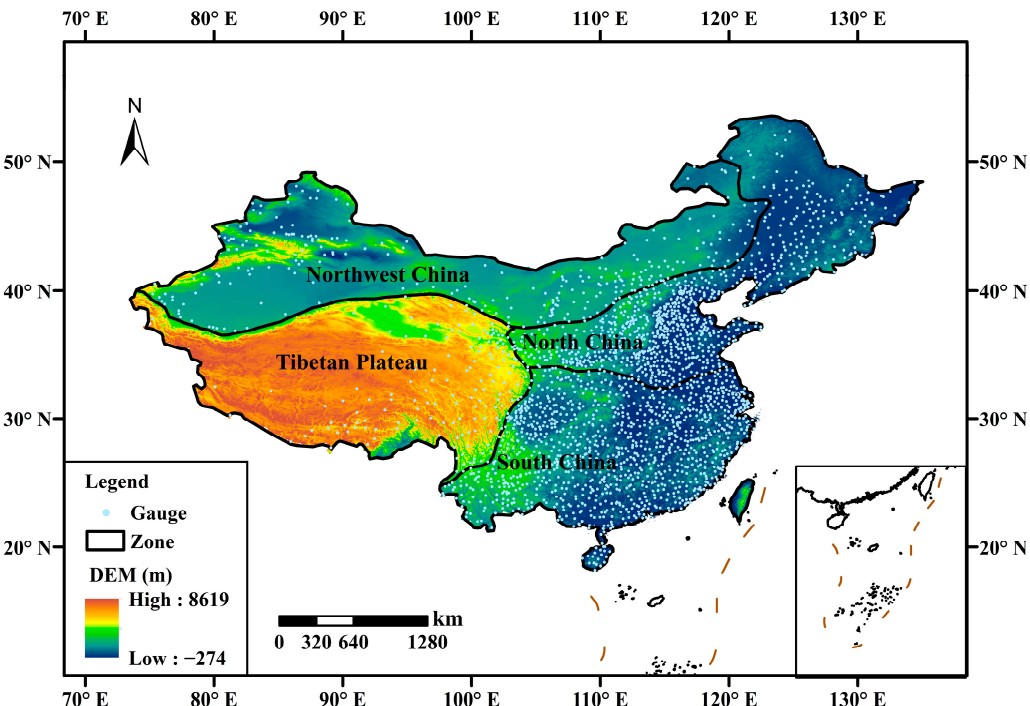

**Figure 1.** Locations of the rain gauges in the four subregions of China.

### 2.2. Datasets

#### 2.2.1. Four SPPs

IMERG is a level 3 product of the GPM mission, which utilizes data from multiple satellite sensors onboard the GPM platform, including information from previous missions such as TRMM. IMERG leverages data from a multitude of Low Earth Orbit (LEO) satellites and complements it with geostationary Earth orbit (GEO) infrared estimates to overcome the limited sampling of individual LEO satellites [13]. IMERG has proven useful in various meteorological and precipitation assessments [49–51]. This study utilized the IMERG v06 Final-Run precipitation dataset, which has a spatial resolution of 0.1° and a temporal resolution of 1 day.

GSMaP is another project under the Japan Precipitation Measurement Mission (PMM) scientific team. The GSMaP algorithm utilizes various Passive Microwave/Infrared

(PMW/IR) sensors, such as the GPM Microwave Imager (GMI) [52]. GSMaP_MVK is obtained by a joint passive microwave and infrared inversion algorithm based on the Kalman filter moving vector method [53]. Another near-real-time product of GSMaP_NRT was developed and attracted many data users because of its short latency (about 3 h after observation). GSMaP_Gauge is then obtained by correcting the GSMaP_MVK using CPC rainfall station data. GSMaP_NRT and GSMaP_Gauge products are used in this study with resolutions of 0.1° and 1 d.

MSWEP is a recently developed global precipitation dataset by Beck et al. [54]. It incorporates global site data, multiple satellite observations, and reanalysis data, and it is revised with some runoff and potential evapotranspiration data. MSWEP has attracted extensive international attention since its release due to its relatively high spatial resolution (0.1°), long time series (1979–2017), and strong data integrity [55]. In this study, MSWEP V2 is adopted in the study, and the daily precipitation was obtained by accumulating precipitation observations for 3 h.

### 2.2.2. Rain Gauge Data

In this study, daily precipitation data from 2344 meteorological gauges in China from 2001 to 2018 were used. The dataset was compiled by the China Meteorological Administration (CMA). The quality of the data set was strictly controlled before release. We preprocessed the missing daily precipitation data using multi-year daily averages at a given point in time. The distribution of rain gauges is shown in Figure 1.

### 2.3. Methods
#### 2.3.1. Conventional Indices

In order to compare the performance of four SPPs generally, we used the point-to-pixel method to calculate AD, RB, RMSE, and Corr between each gauge's observation data and satellite precipitation data, and carried out the spatial average of the index values of different sub-regions. The calculation methods are as follows [56,57]:

$$AD = \frac{1}{N} \sum_{i=1}^{N} |S_i - G_i|, \tag{1}$$

$$RB = \frac{\sum_{i=1}^{N} (S_i - G_i)}{\sum_{i=1}^{N} (G_i)} * 100\%, \tag{2}$$

$$RMSE = \sqrt{\frac{1}{N} \sum_{i=1}^{N} (S_i - G_i)^2}, \tag{3}$$

$$Corr = \frac{\sum_{i=1}^{N} (S_i - \overline{S})(G_i - \overline{G})}{\sqrt{\sum_{i=1}^{N} (S_i - \overline{S})^2} \sqrt{\sum_{i=1}^{N} (G_i - \overline{G})^2}}, \tag{4}$$

where $N$ is the number of gauges; $S$ and $G$ are the SPPs data and the gauge observations, respectively; $\overline{S}$ is the average of the SPPs data; and $\overline{G}$ is the average of the gauge observations.

#### 2.3.2. Spatial Correlation Analysis
BMI

The BMI, which encompasses both global and local variations, has been widely acknowledged as a suitable tool to compare the spatial correlation between two geographic elements [58]. In this research, we employed the BMI method to investigate the spatial correlation between SPPs and gauge observations.

The calculation equations of global BMI and local BMI can be expressed as follows:

$$I_B = \frac{N \sum_i^N \sum_{j \neq i}^N W_{ij} Z_i^G Z_j^S}{(N-1) \sum_i^N \sum_{j \neq i}^N W_{ij}} \tag{5}$$

$$I_i^B = Z_i^G \sum_{j=1}^N W_{ij} Z_j^S, \tag{6}$$

where $I_B$, $I_i^B$ refer to the global and local BMI, respectively; $N$ is the total number of gauges; $Z_i^G$ and $Z_j^S$ refer to the standardized value of gauge observations for $i$ site and the standardized value of SPPs for the j site, respectively; and $W_{ij}$ is the Euclidean distance weight between i and j sites.

The values of $I_B$, $I_i^B$ range from −1 to 1. A positive value indicates a positive spatial correlation between SPPs and gauge observations, while a negative value indicates a negative spatial correlation [59]. $I_B$ can also be expressed as the slope of the linear fit to the bivariate Moran scatter plot, which consists of a plot with the spatially lagged standardized satellite precipitation data on the y-axis and the standardized gauge observations on the x-axis.

The LISA cluster map is derived from local BMI, and identifies the spatial correlation clusters as: High-High (H-H), Low-Low (L-L), High-Low (H-L), and Low-High (L-H). H-H and L-L clusters indicate that there is a positive correlation between SPPs and satellite gauge observations in this region, while H-L and L-H clusters indicate that there is a negative correlation between them. In this study, the significance of the local BMI was assessed by a permutation test [58], and the significance level was set to 0.05.

Selected Precipitation Indices

The spatial correlation between gauge observations and SPPs was achieved by calculating the BMI between precipitation indices computed using gauge observations and those obtained from SPPs. Eight widely used precipitation indicators defined by the Expert Group on Climate Change Detection and Indices (ETCCDI) [60,61] were selected. The six indicators were mainly divided into three categories, and their definitions are shown in Table 1.

**Table 1.** Detailed information on precipitation indices.

| Sort | Index | Definition | Units |
|---|---|---|---|
| Total indices | ATP | Annual total precipitation | mm |
| | ATD | Annual total precipitation days | days |
| Persistent indices | CDD | Maximum number of consecutive dry days | days |
| | CWD | Maximum number of consecutive wet days | days |
| Extreme indices | R95 | The 95th percentile of daily precipitation on wet days | mm |
| | Rmax | Annual max 1-day precipitation | mm |
| Frequency indices | R25 | Annual count of days when daily precipitation is >25 mm | days |
| | R50 | Annual count of days when daily precipitation is >50 mm | days |

## 3. Results

### 3.1. Conventional Indices

The conventional indices of the satellite precipitation data in China and its subregions were presented in Table 2. The results showed that MSWEP had the best performance, with a Corr of 0.78, an AD of 1.6, a RB of −5%, and a RMSE of 5. IMERG was ranked second, while GSMaP_NRT performed the poorest due to the lack of merged rain gauge data. Among the four regions, the highest correlation between SPPs and gauge observations was observed in SC, whereas the lowest correlation was in NWC. MSWEP show the highest correlation in all four regions, with a Corr of 0.80, 0.78, 0.70, and 0.74 respectively. GSMaP_Gauge and GSMaP_NRT largely underestimated the daily precipitation in SC, but overestimated it in NC, NWC, and TP, particularly in NWC, with an RB of 53% and 209%, respectively. IMERG overestimated the daily precipitation in all four regions, with a RB of 10%, 6%, 22%, and 13%, respectively. MSWEP mainly underestimated daily precipitation in NC, SC, and NWC by −8%, −6%, and −3%, respectively, but showed a positive deviation

of 18% in TP. Overall, MSWEP and IMERG performed better than GSMaP_Gauge and GSMaP_NRT in four sub-regions, especially in TP and NWC.

**Table 2.** The conventional indices of the satellite precipitation data in China and its sub-regions.

| Index | SPP | NC | SC | NWC | TP | China |
|---|---|---|---|---|---|---|
| Corr | GSMaP_Gauge | 0.62 | 0.65 | 0.44 | 0.55 | 0.61 |
| | GSMaP_NRT | 0.49 | 0.60 | 0.37 | 0.47 | 0.53 |
| | IMERG | 0.73 | 0.73 | 0.64 | 0.66 | 0.71 |
| | MSWEP | 0.80 | 0.78 | 0.70 | 0.74 | 0.78 |
| AD | GSMaP_Gauge | 1.5 | 3.1 | 0.5 | 1.5 | 2.2 |
| | GSMaP_NRT | 1.9 | 3.1 | 2.1 | 1.8 | 2.4 |
| | IMERG | 1.4 | 3.0 | 0.2 | 1.3 | 2.0 |
| | MSWEP | 1.0 | 2.4 | 0.0 | 1.1 | 1.6 |
| RB | GSMaP_Gauge | 4% | −8% | 53% | 23% | 5% |
| | GSMaP_NRT | 40% | −12% | 209% | 56% | 35% |
| | IMERG | 10% | 6% | 22% | 13% | 9% |
| | MSWEP | −8% | −6% | −3% | 18% | −5% |
| RMSE | GSMaP_Gauge | 5.4 | 8.7 | 3.2 | 4.2 | 6.7 |
| | GSMaP_NRT | 8.0 | 9.6 | 6.4 | 5.5 | 8.4 |
| | IMERG | 4.7 | 7.9 | 2.2 | 3.5 | 5.9 |
| | MSWEP | 3.9 | 6.7 | 2.0 | 3.0 | 5.0 |

The high performance of the MSWEP product can be attributed to its integration of precipitation estimates from multiple sources, including satellite-based estimates, gauge-based observations, and reanalysis data. Notably, the incorporation of a large number of gauge observations has significantly enhanced its accuracy, which may explain why it has the best performance in the study. Moreover, SC has the highest density of rain gauges on the Chinese mainland. This provides ample data sources for the SPPs to improve their estimates, especially for the MSWEP and GSMaP_Gauge. In contrast, NWC has a sparse rain gauge network, which restricts the improvement of SPPs in this region. This difference in rain gauge density may be one of the reasons why SC has a higher correlation with gauge observations compared to NWC.

*3.2. Spatial Correlation Analysis*

3.2.1. Total Indices

BMI in China

The global BMI of ATP and ATD was calculated and their bivariate Moran scatter plots were shown in Figures 2 and 3. As illustrated in Figure 2, the BMI of ATP for GSMaP_Gauge, GSMaP_NRT, IMERG, and MSWEP are 0.93, 0.74, 0.96, and 0.95, respectively. Products such as GSMaP_Gauge, IMERG, and MSWEP demonstrate good spatial correlation with the gauge observations and effectively capture the spatial pattern of ATP. Meanwhile, GSMaP_NRT exhibits a significantly lower BMI compared to the other products. In Figure 3, the BMI of ATD for the four products were found to be 0.77, 0.71, 0.68, and 0.89, respectively. Despite having the best spatial correlation for annual precipitation, IMERG demonstrates poor performance in capturing annual precipitation days. However, the scatter plot in the figure indicates that MSWEP can effectively capture not only the spatial distribution characteristics of annual precipitation but also the number of annual precipitation days.

The LISA cluster maps of ATP and ATD are presented in Figures 4 and 5. Figure 4 indicates that the ATP produced by GSMaP_Gauge, IMERG, MSWEP, and the gauge observations demonstrate a strong positive spatial correlation with L-L clusters in NWC, NC and TP, and H-H clusters in central and southern SC. Meanwhile, there is a notable negative spatial correlation between the gauge observations and GSMaP_NRT in the TP, which is evidenced by the L-H clusters in the eastern region of the plateau, suggesting that GSMaP_NRT significantly overestimates ATP.

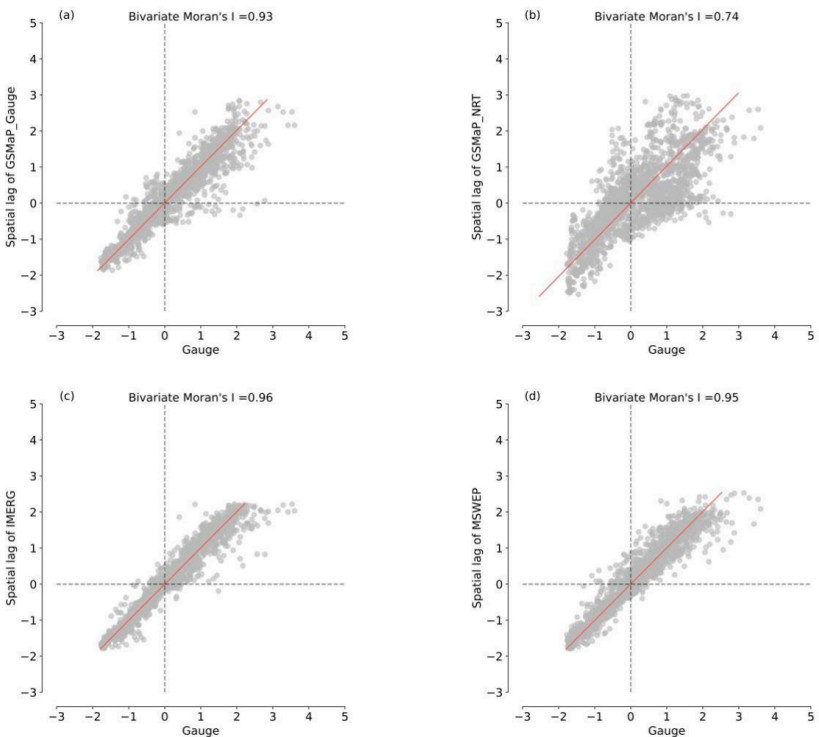

**Figure 2.** The global BMI scatter plot of ATP between the gauge observations and the four SPPs: (**a**) GSMaP_Gauge, (**b**) GSMaP_NRT, (**c**) IMERG, and (**d**) MSWEP.

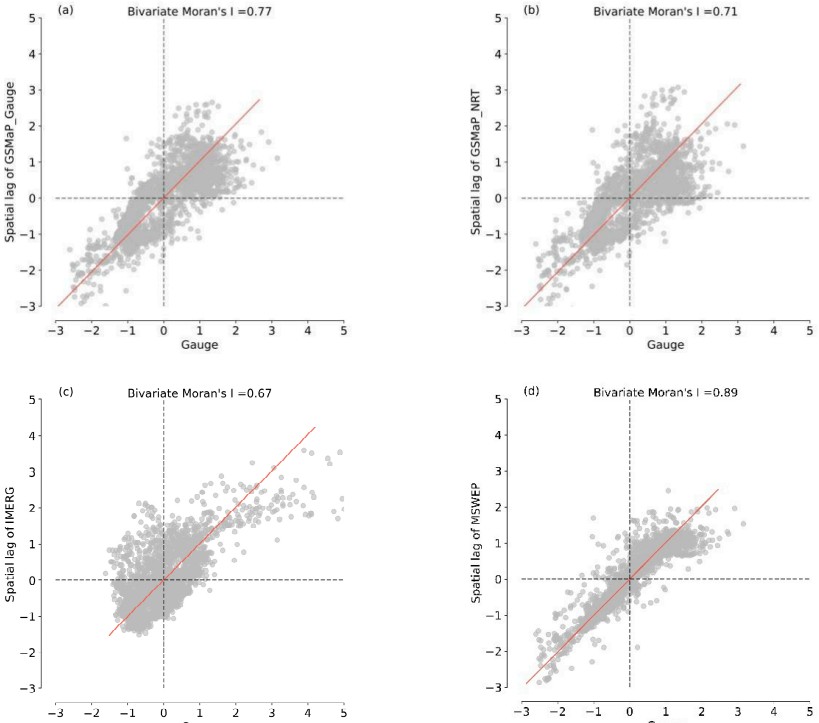

**Figure 3.** The global BMI scatter plot of ATD between the gauge observations and the four SPPs: (**a**) GSMaP_Gauge, (**b**) GSMaP_NRT, (**c**) IMERG, and (**d**) MSWEP.

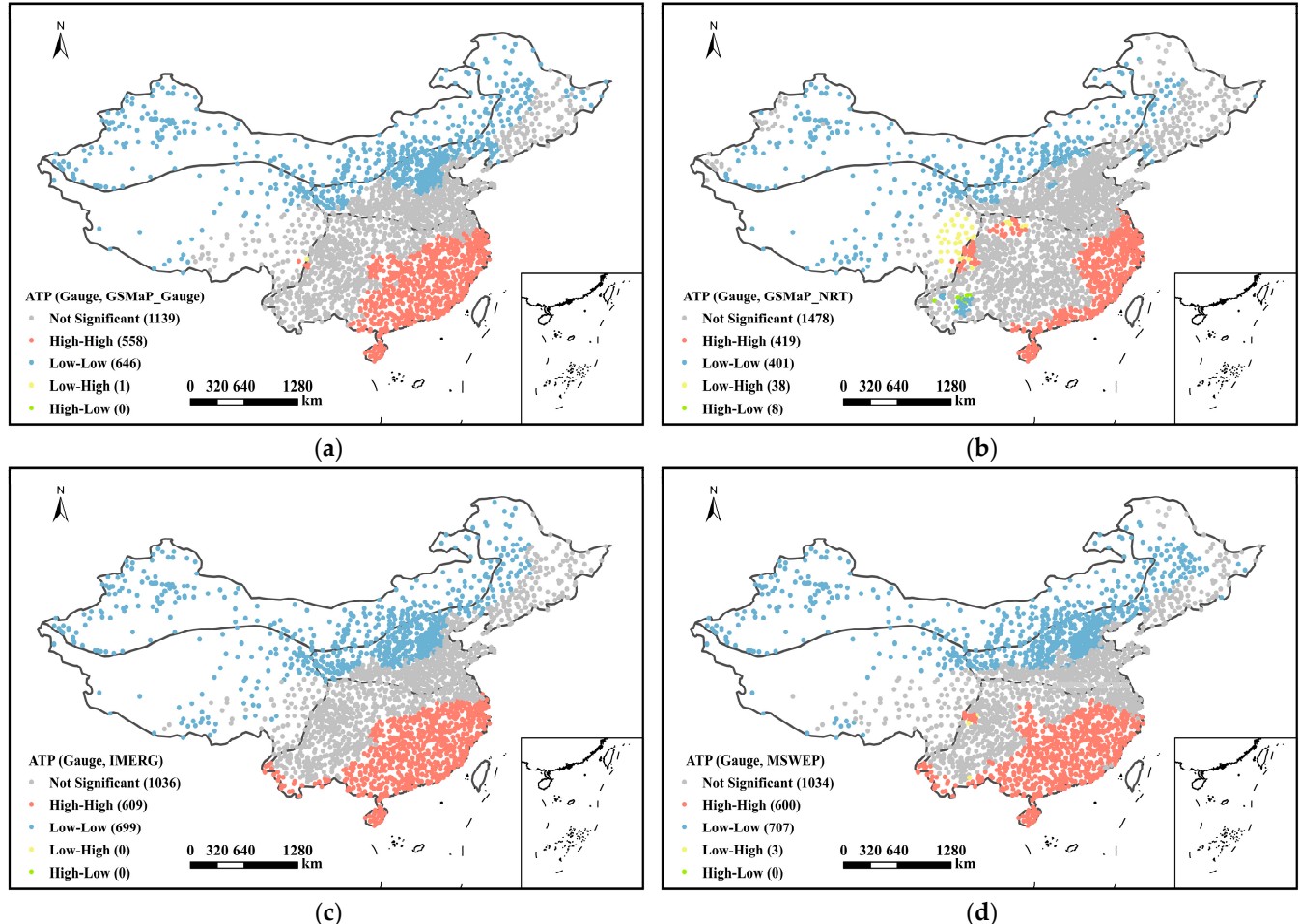

**Figure 4.** The LISA cluster maps of ATP between the gauge observations and the four SPPs and the number in the bracket represent the corresponding number of gauges: (**a**) GSMaP_Gauge, (**b**) GSMaP_NRT, (**c**) IMERG, and (**d**) MSWEP.

In Figure 5, the ATD obtained by the four SPPs and the gauge observations display significant positive spatial correlation in NWC and NC with L-L clusters. GSMaP_NRT and GSMaP_Gauge show a strong positive correlation in the northeast and northwest regions of SC and the eastern region of the TP, as indicated by H-H clusters. IMERG demonstrates strong positive correlation with H-H clusters in the eastern SC, but it also shows a negative spatial correlation with L-H clusters in the junction between NC and SC, suggesting that IMERG significantly overestimates ATD. MSWEP demonstrates a higher proportion of H-H clusters in the SC and TP compared to the other precipitation products.

BMI in Four Sub-Regions

The BMI for the total indices of four sub-regions were calculated and presented in Table 3. Results indicate that IMERG has the highest spatial correlation with the gauge observations for ATP across all sub-regions. Among the ATD, MSWEP performed the best in both NC and SC. In summary, MSWEP excels at capturing the spatial distribution of total indices in NC and SC, IMERG performs well in NWC, and both IMERG and GSMaP_Gauge perform well in Tibet Plateau TP.

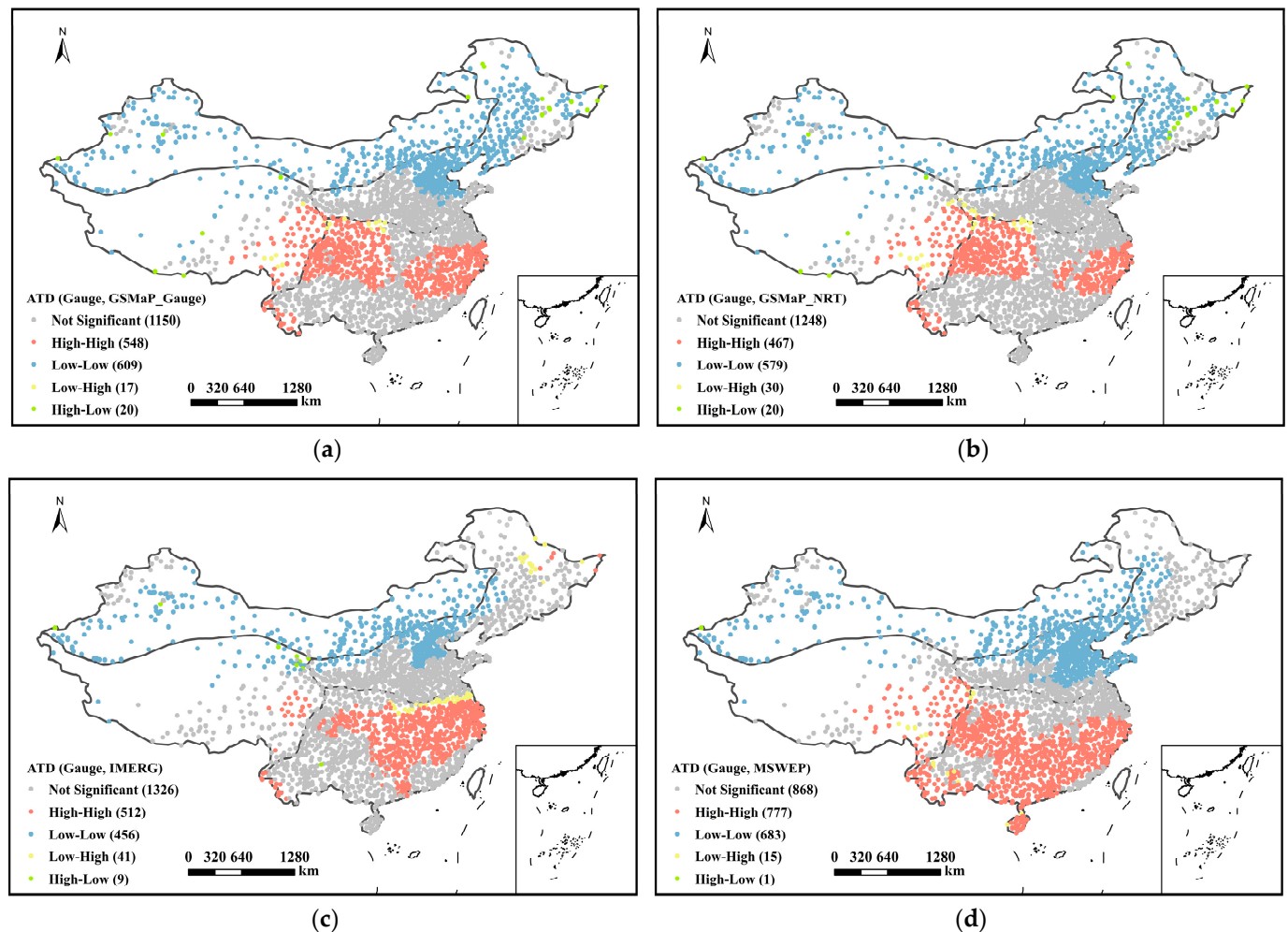

**Figure 5.** The LISA cluster maps of ATD between the gauge observations and the four SPPs and the number in the bracket represent the corresponding number of gauges: (**a**) GSMaP_Gauge, (**b**) GSMaP_NRT, (**c**) IMERG, and (**d**) MSWEP.

**Table 3.** The global BMI of total indices in four sub-regions.

| Index | Sub-Region | GSMaP_Gauge | GSMaP_NRT | IMERG | MSWEP |
|:---:|:---:|:---:|:---:|:---:|:---:|
| | NC | 0.79 | 0.65 | 0.88 | 0.85 |
| | SC | 0.80 | 0.45 | 0.87 | 0.83 |
| ATP | NWC | 0.65 | 0.38 | 0.73 | 0.65 |
| | TP | 0.67 | 0.63 | 0.67 | 0.60 |
| | NC | 0.38 | 0.35 | 0.62 | 0.84 |
| | SC | 0.30 | 0.20 | 0.01 | 0.65 |
| ATD | NWC | 0.53 | 0.51 | 0.63 | 0.60 |
| | TP | 0.60 | 0.60 | 0.60 | 0.60 |

### 3.2.2. Persistent Indices

### BMI in China

The results of the global BMI and its scatter plots for CDD and CWD between SPPs and gauge observations are presented in Figures 6 and 7, respectively. As can be seen from the scatter plots, the global BMI for the CDD of GSMaP_Gauge, GSMaP_NRT, IMERG, and MSWEP are 0.66, 0.65, 0.67, and 0.78, respectively. The global BMI for the CWD of the four products were 0.73, 0.65, 0.70, and 0.78, respectively. Among the four products, MSWEP has the highest spatial correlation with the gauge observations, indicating that it has the best performance in capturing CDD and CWD. Conversely, GSMaP_NRT, which did not incorporate gauge observations, performed the worst.

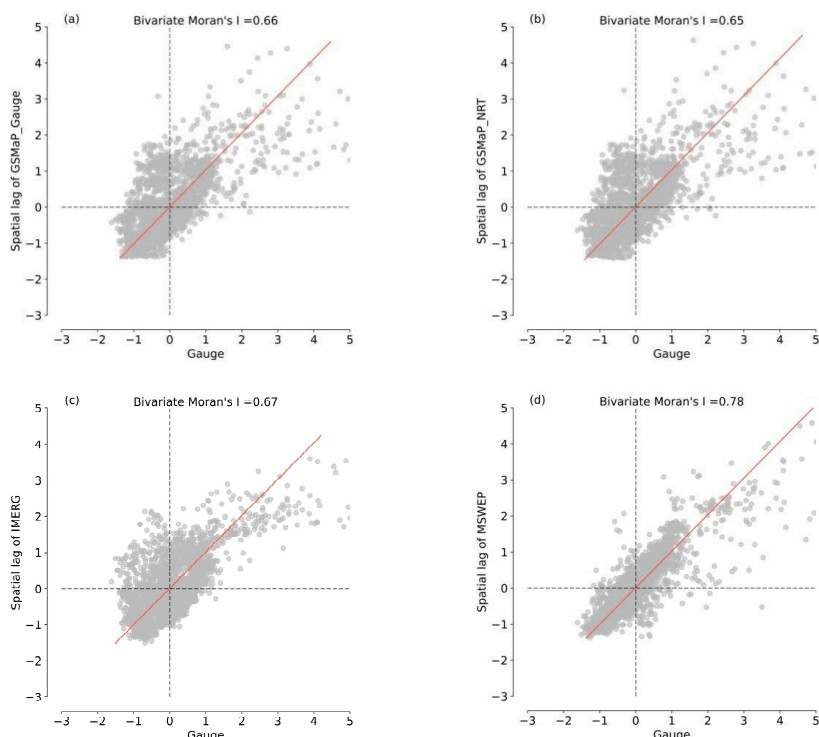

**Figure 6.** The global BMI scatter plot of CDD between the gauge observations and the four SPPs: (**a**) GSMaP_Gauge, (**b**) GSMaP_NRT, (**c**) IMERG, and (**d**) MSWEP.

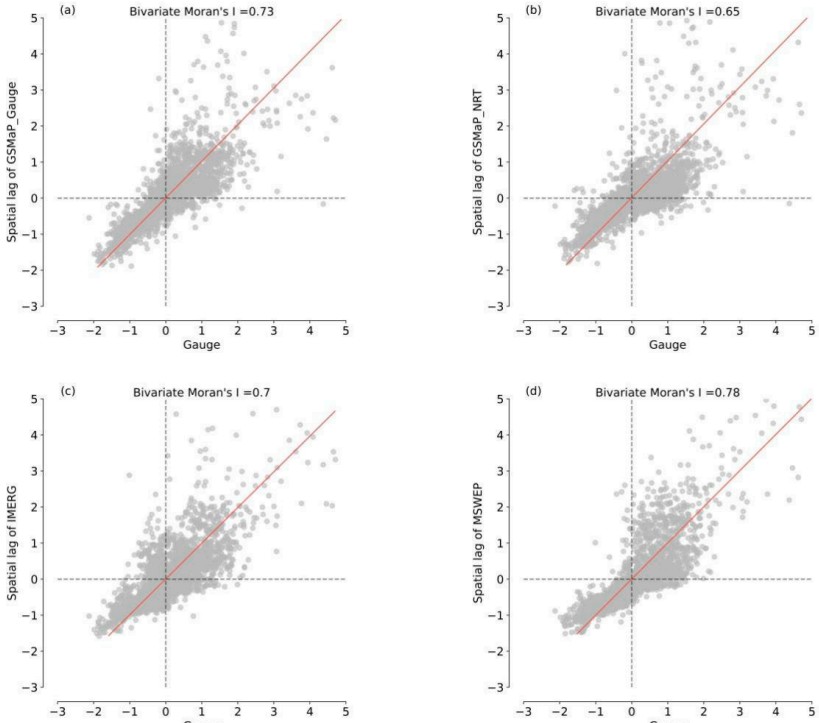

**Figure 7.** The global BMI scatter plot of CWD between the gauge observations and the four SPPs: (**a**) GSMaP_Gauge, (**b**) GSMaP_NRT, (**c**) IMERG, and (**d**) MSWEP.

The LISA cluster maps for CDD and CWD are depicted in Figures 8 and 9, respectively. It is evident from Figure 8 that the CDD values obtained from the four SPPs exhibit a substantial positive spatial correlation with the gauge observations, as demonstrated by the L-L clusters in the northern region of SC and the H-H clusters in NWC and TP. However,

a significant negative correlation between the GSMaP_NRT and GSMaP_Gauge and the gauge observations is observed in the northeast of NC, indicated by the L-H clusters. IMERG exhibits a negative spatial correlation with H-L clusters in the southeast of NC and L-H clusters in the west and south of SC. MSWEP displays a negative spatial correlation with H-L clusters in the border region between SC and TP.

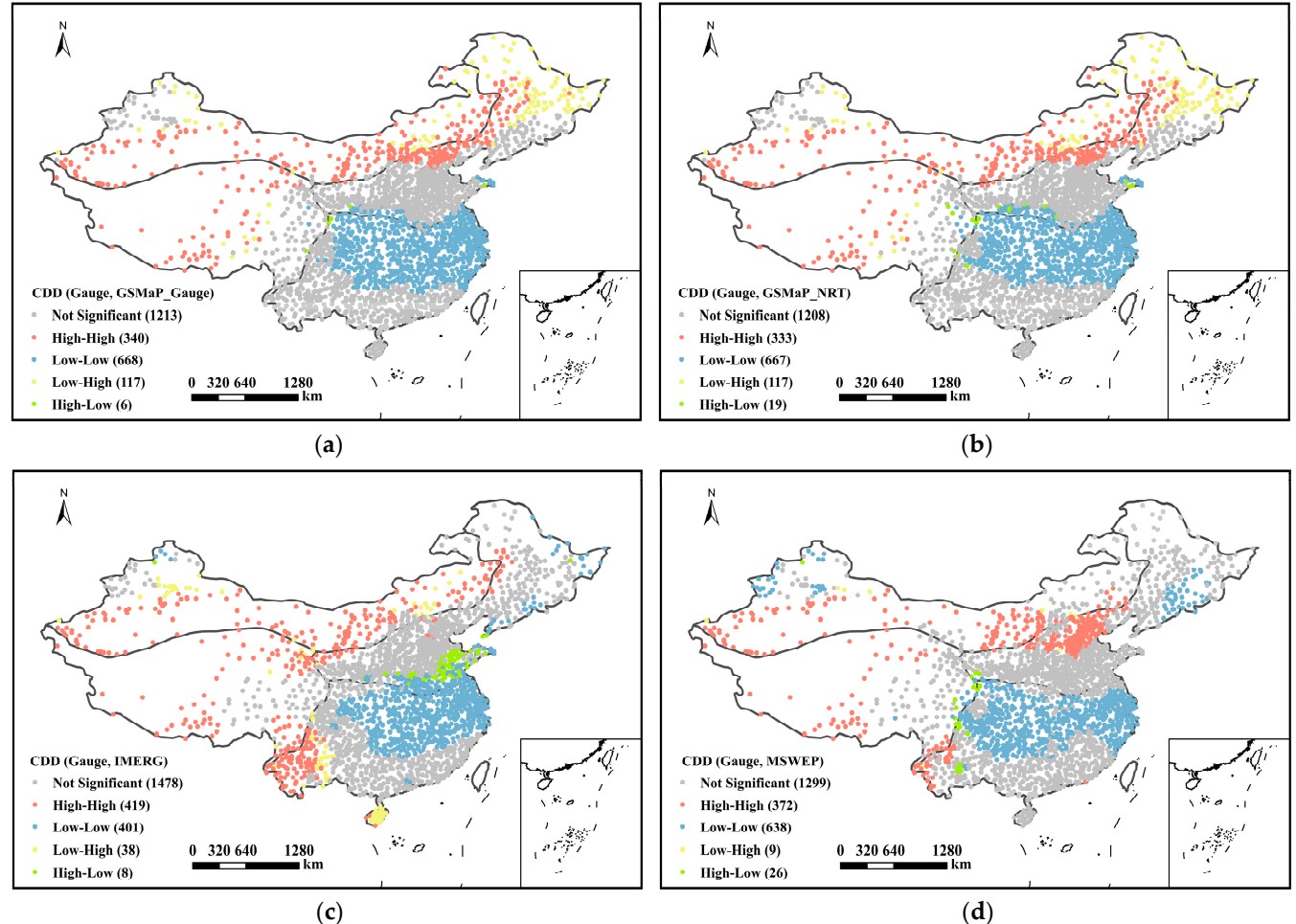

**Figure 8.** The LISA cluster maps of CDD between the gauge observations and the four SPPs and the number in the bracket signify represent the corresponding number of gauges: (**a**) GSMaP_Gauge, (**b**) GSMaP_NRT, (**c**) IMERG, and (**d**) MSWEP.

In Figure 9, the CWD values obtained from the four SPPs exhibit a positive spatial correlation with the gauge observations, as demonstrated by the L-L clusters in NWC and NC as well as the H-H clusters in TP and the southern region of SC. The L-H and H-L clusters, which are present in the results of GSMaP_NRT, GSMaP_Gauge, and IMERG, are much smaller compared to those in the CDD values. This indicates that the spatial correlations for CWD are much better than those for CDD.

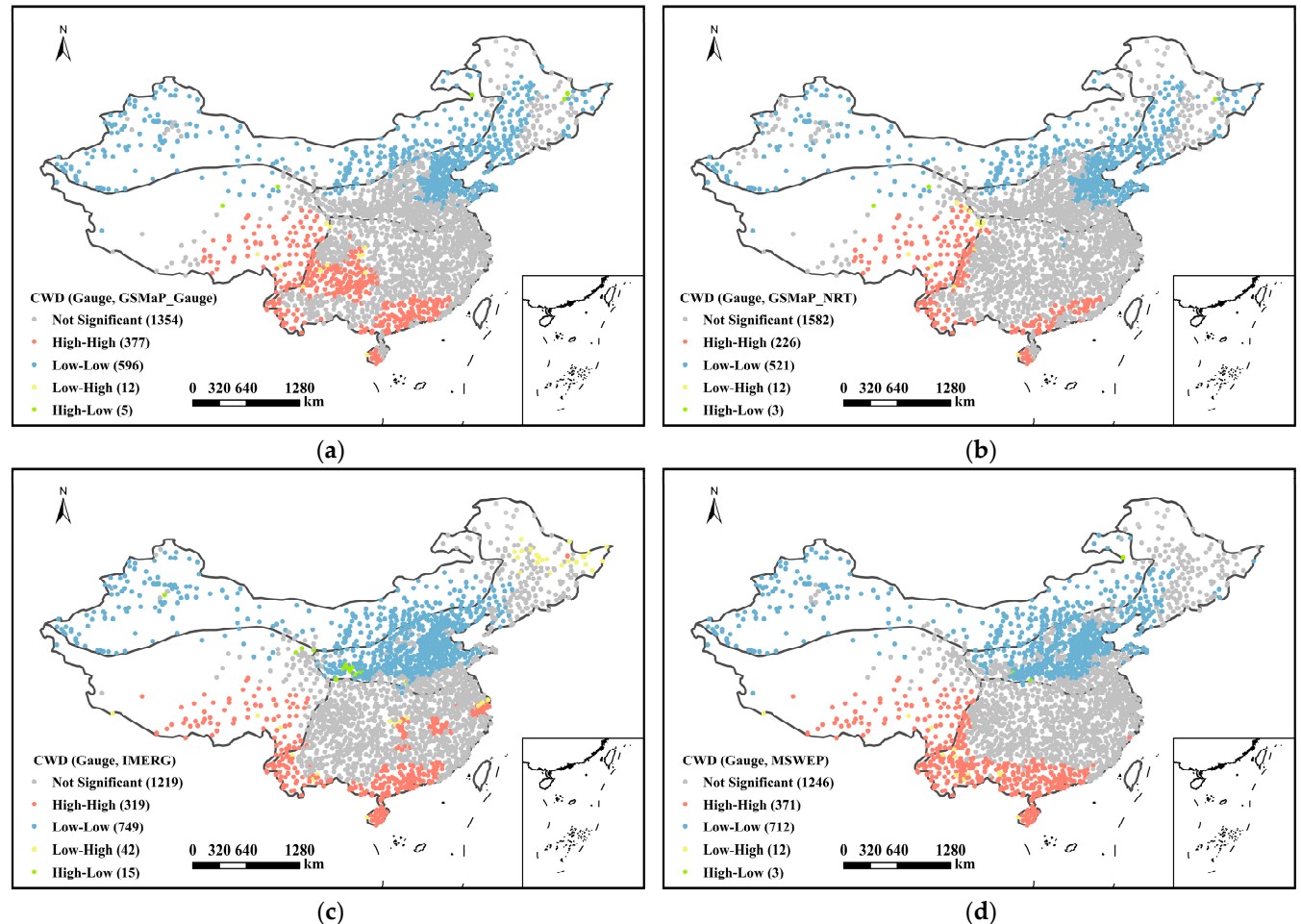

**Figure 9.** The LISA cluster maps of CWD between the gauge observations and the four SPPs and the number in the bracket represent the corresponding number of gauges: (**a**) GSMaP_Gauge, (**b**) GSMaP_NRT, (**c**) IMERG, and (**d**) MSWEP.

BMI in Four Sub-Regions

The results of the global BMI analysis of the persistent indices in four regions are presented in Table 4. The analysis reveals that MSWEP has the highest spatial correlation with CDD and CWD in all four regions. In comparison, the CDD values obtained from GSMaP_Gauge, GSMaP_NRT, and IMERG have significantly lower spatial correlations with the gauge observations in NC and SC when compared to MSWEP. Furthermore, the CWD values obtained from the four SPPs exhibit the worst spatial correlation in NWC. Among the four SPPs, IMERG shows the weakest performance in NC and SC.

**Table 4.** The global BMI of persistent indices in four sub-regions.

| Index | Sub-Region | GSMaP_Gauge | GSMaP_NRT | IMERG | MSWEP |
|-------|------------|-------------|-----------|-------|-------|
| CDD | NC | 0.20 | 0.18 | 0.59 | 0.85 |
| | SC | 0.56 | 0.56 | 0.54 | 0.71 |
| | NWC | 0.44 | 0.41 | 0.60 | 0.63 |
| | TP | 0.58 | 0.58 | 0.56 | 0.67 |
| CWD | NC | 0.60 | 0.60 | 0.32 | 0.56 |
| | SC | 0.51 | 0.54 | 0.49 | 0.62 |
| | NWC | 0.35 | 0.33 | 0.39 | 0.40 |
| | TP | 0.49 | 0.45 | 0.47 | 0.59 |

### 3.2.3. Extreme Indices

BMI in China

The results of the global BMI and its scatter plots for R95 and Rmax between SPPs and gauge observations are presented in Figures 10 and 11. The results reveal that the global BMI values of R95 for GSMaP_Gauge, GSMaP_NRT, IMERG, and MSWEP are 0.83, 0.73, 0.84, and 0.87, respectively. Similarly, the global BMI values of Rmax for the four products are 0.83, 0.46, 0.91, and 0.88, respectively.

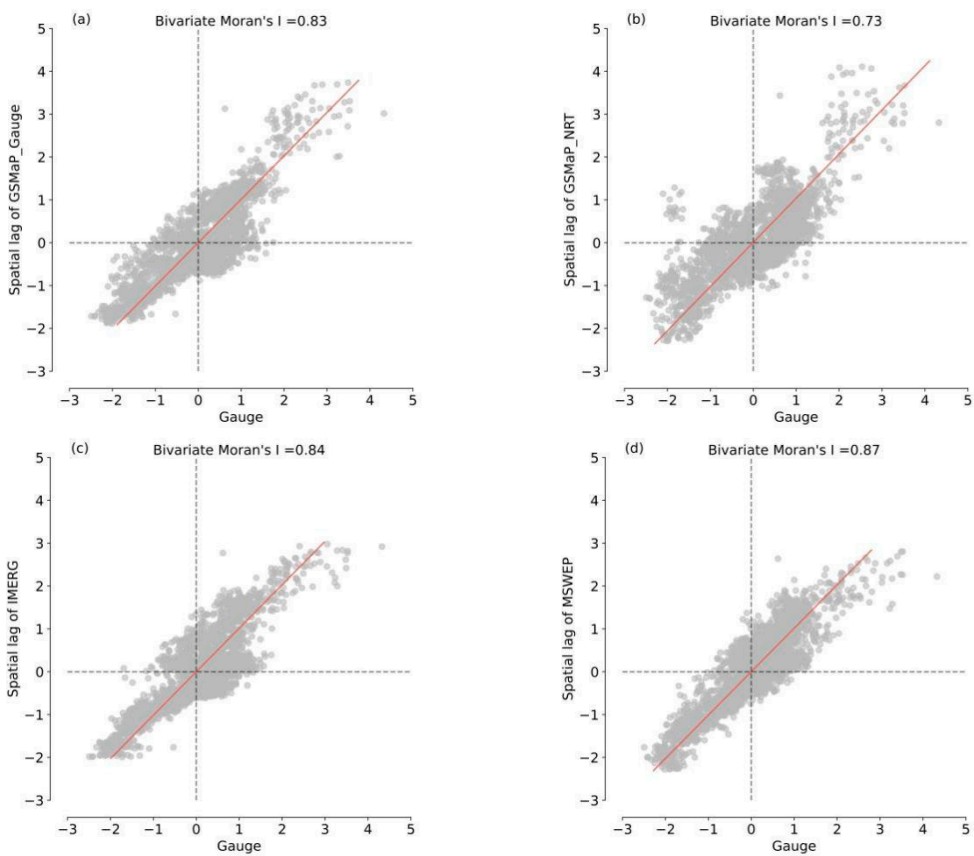

**Figure 10.** The global BMI scatter plot of R95 between the gauge observations and the four SPPs: (**a**) GSMaP_Gauge, (**b**) GSMaP_NRT, (**c**) IMERG and (**d**) MSWEP.

The results indicate that the GSMaP_Gauge, IMERG, and MSWEP accurately represent the spatial distribution of extreme events. Among these, MSWEP shows the best performance in R95, while IMERG is best in Rmax. In contrast, GSMaP_NRT demonstrates the weakest performance among the four SPPs.

The LISA cluster maps of R95 and Rmax are shown in Figures 12 and 13. The results indicate that the R95 and Rmax obtained from the four SPPs display significant positive spatial correlations with the gauge observations, as evidenced by the presence of L-L clusters in the NWC and TP. Furthermore, GSMaP_Gauge, IMERG, and MSWEP exhibit significant H-H clusters in the central and southern SC. However, GSMaP_NRT only exhibits significant H-H clusters in the coastal areas of SC. The worst performing product, GSMaP_NRT, shows significant negative spatial correlation with H-L clusters in central SC and L-H clusters in the western NC and eastern NWC.

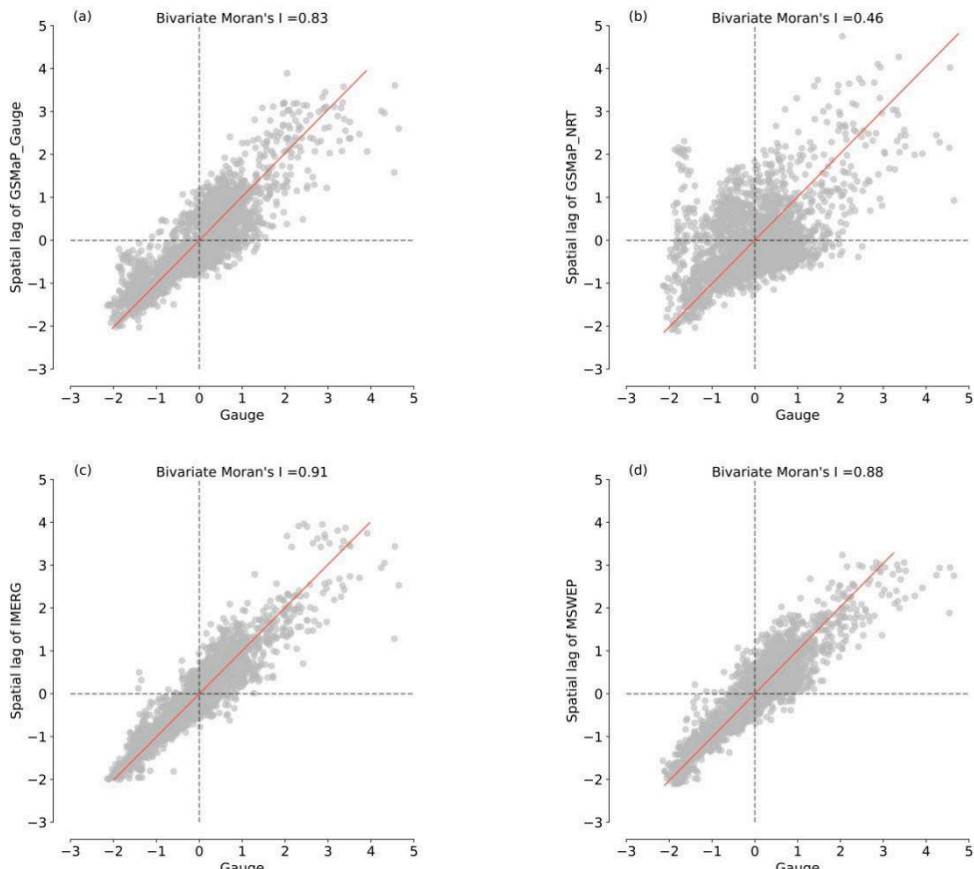

**Figure 11.** The global BMI scatter plot of Rmax between the gauge observations and the four SPPs: (**a**) GSMaP_Gauge, (**b**) GSMaP_NRT, (**c**) IMERG, and (**d**) MSWEP.

BMI in Four Sub-Regions

The results of the BMI of extreme indices in the four sub-regions are shown in Table 5. The analysis shows that IMERG has the highest positive spatial correlation with gauge observations in all four sub-regions, as reflected in the highest values of the global BMI. Meanwhile, the global BMI of GSMaP_NRT was significantly lower in comparison to the other precipitation products, particularly in NWC, where the values of global BMI were only 0.12 and −0.12 for R95 and Rmax, respectively. The results also indicate that the spatial correlation of extreme indices is weaker in TP compared to other regions.

**Table 5.** The global BMI of extreme indices in four sub-regions.

| Index | Sub-Region | GSMaP_Gauge | GSMaP_NRT | IMERG | MSWEP |
|-------|------------|-------------|-----------|-------|-------|
| R95 | NC | 0.67 | 0.59 | 0.83 | 0.83 |
| | SC | 0.80 | 0.70 | 0.73 | 0.74 |
| | NWC | 0.67 | 0.12 | 0.83 | 0.71 |
| | TP | 0.53 | 0.43 | 0.69 | 0.57 |
| Rmax | NC | 0.69 | 0.15 | 0.83 | 0.81 |
| | SC | 0.71 | 0.47 | 0.80 | 0.74 |
| | NWC | 0.25 | −0.12 | 0.84 | 0.73 |
| | TP | 0.52 | 0.38 | 0.57 | 0.47 |

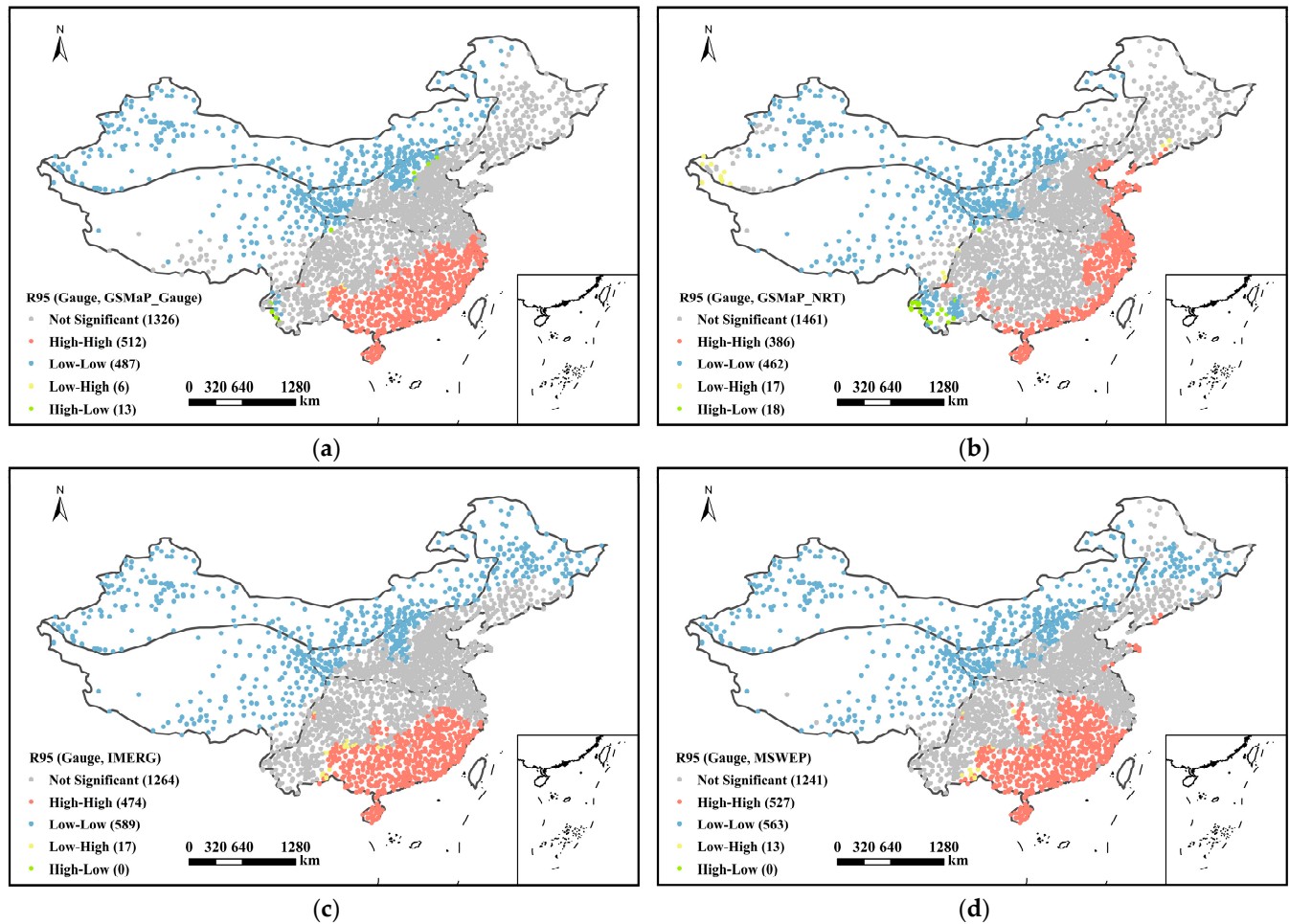

**Figure 12.** The LISA cluster maps of R95 between the gauge observations and the four SPPs and the number in the bracket represent the corresponding number of gauges: (**a**) GSMaP_Gauge, (**b**) GSMaP_NRT, (**c**) IMERG, and (**d**) MSWEP.

### 3.2.4. Frequency Indices

BMI in China

The results of the global BMI and its scatter plots for R25 and R50 between SPPs and gauge observations are presented in Figures 14 and 15. The results reveal that the global BMI values of R25 for GSMaP_Gauge, GSMaP_NRT, IMERG, and MSWEP are 0.92, 0.77, 0.96, and 0.94, respectively. Similarly, the global BMI values of R50 for the four products are 0.89, 0.73, 0.92, and 0.88, respectively.

The results suggest that IMERG performs the best among the four selected products, indicating a robust ability to detect extreme precipitation events. GSMaP_Gauge and MSWEP also demonstrate a strong ability in detecting extreme precipitation events. However, GSMaP_NRT shows the weakest performance among the four products. It is important to note that R50 is temporally non-stationary in some regions due to high thresholds, which will lead to the uncertainties in BMI results.

The LISA cluster maps of R25 and R50 are shown in Figures 16 and 17. The results suggest that the R25 and R50 derived from the four SPPs exhibit considerable positive spatial correlations with the gauge observations, with the H-H cluster located in the eastern SC and L-L clusters located in the NWC. Conversely, the GSMaP_NRT results show noteworthy negative spatial correlations with L-H clusters in the eastern TP and H-L clusters in southwest SC. The GSMaP_Gauge also indicates a slight presence of L-H clusters in eastern TP.

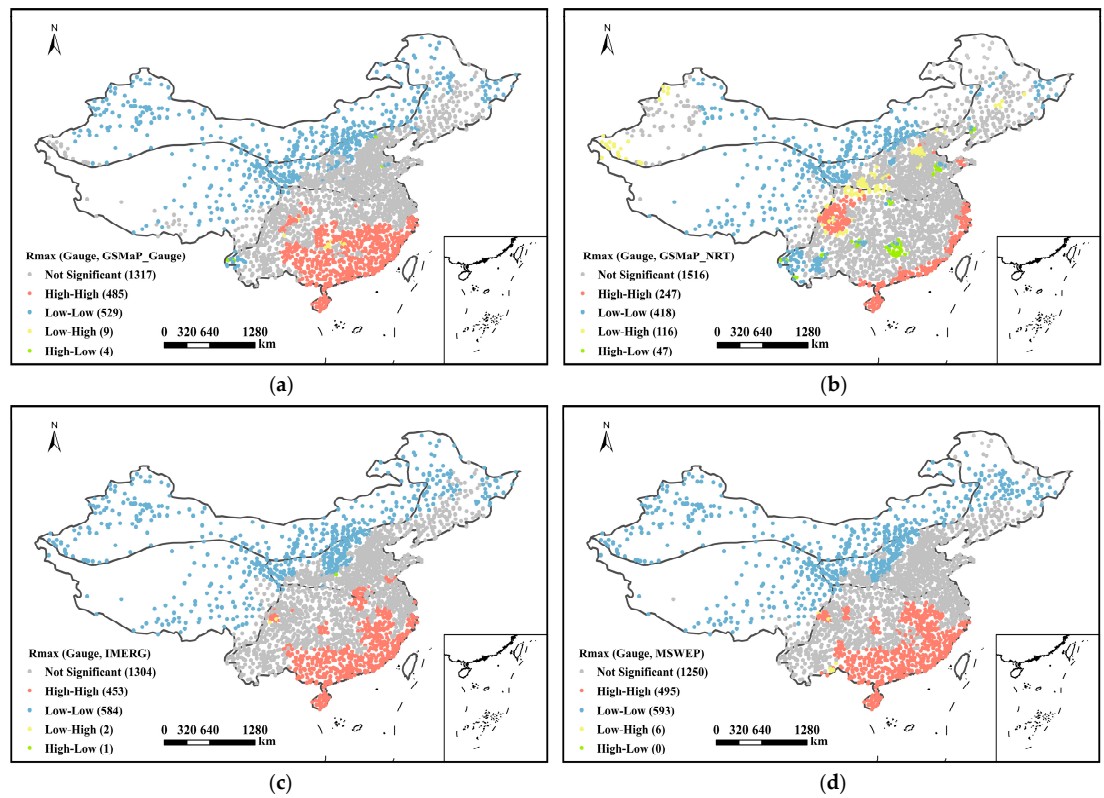

**Figure 13.** The LISA cluster maps of Rmax between the gauge observations and the four SPPs and the number in the bracket represent the corresponding number of gauges: (**a**) GSMaP_Gauge, (**b**) GSMaP_NRT, (**c**) IMERG, and (**d**) MSWEP.

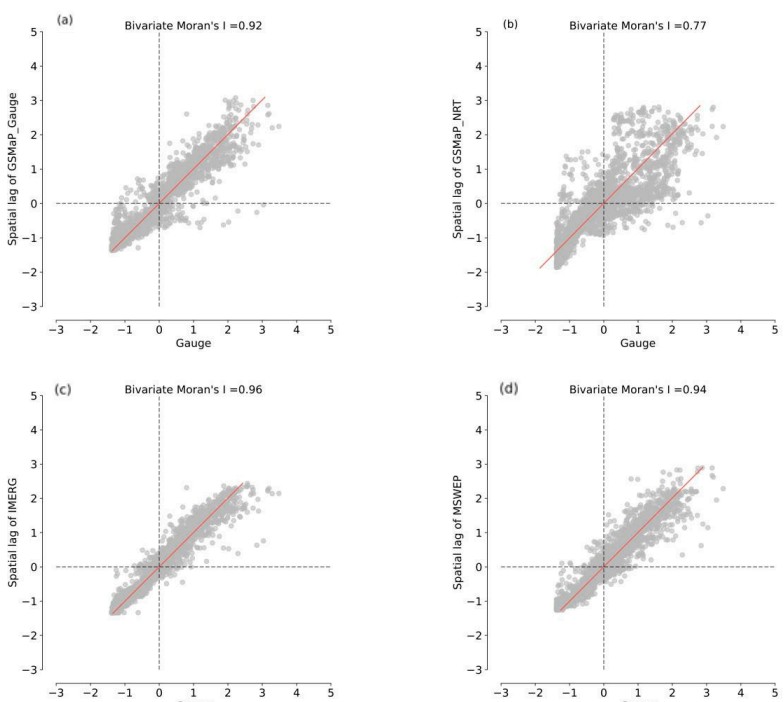

**Figure 14.** The global BMI scatter plot of R25 between the gauge observations and the four SPPs: (**a**) GSMaP_Gauge, (**b**) GSMaP_NRT, (**c**) IMERG, and (**d**) MSWEP.

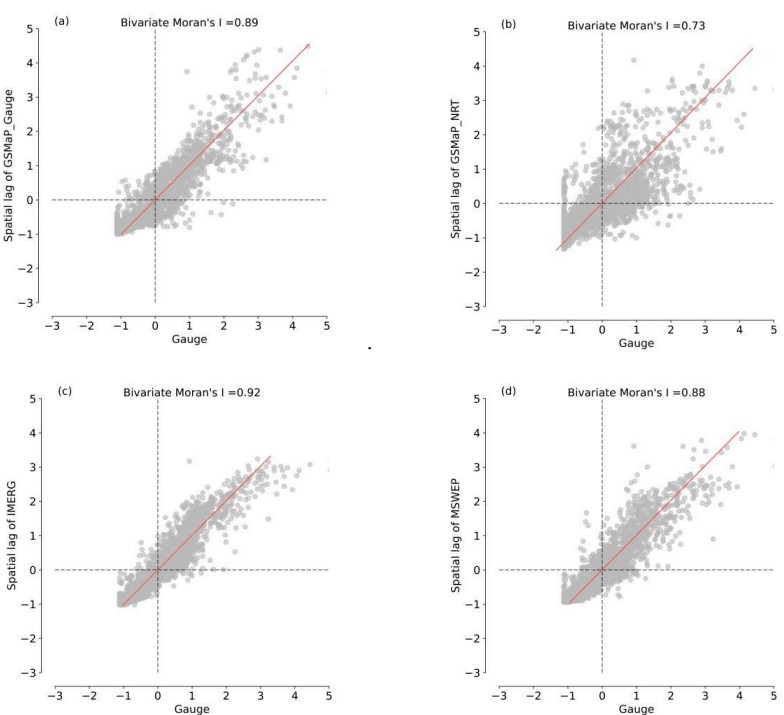

**Figure 15.** The global BMI scatter plot of R50 between the gauge observations and the four SPPs: (**a**) GSMaP_Gauge, (**b**) GSMaP_NRT, (**c**) IMERG, and (**d**) MSWEP.

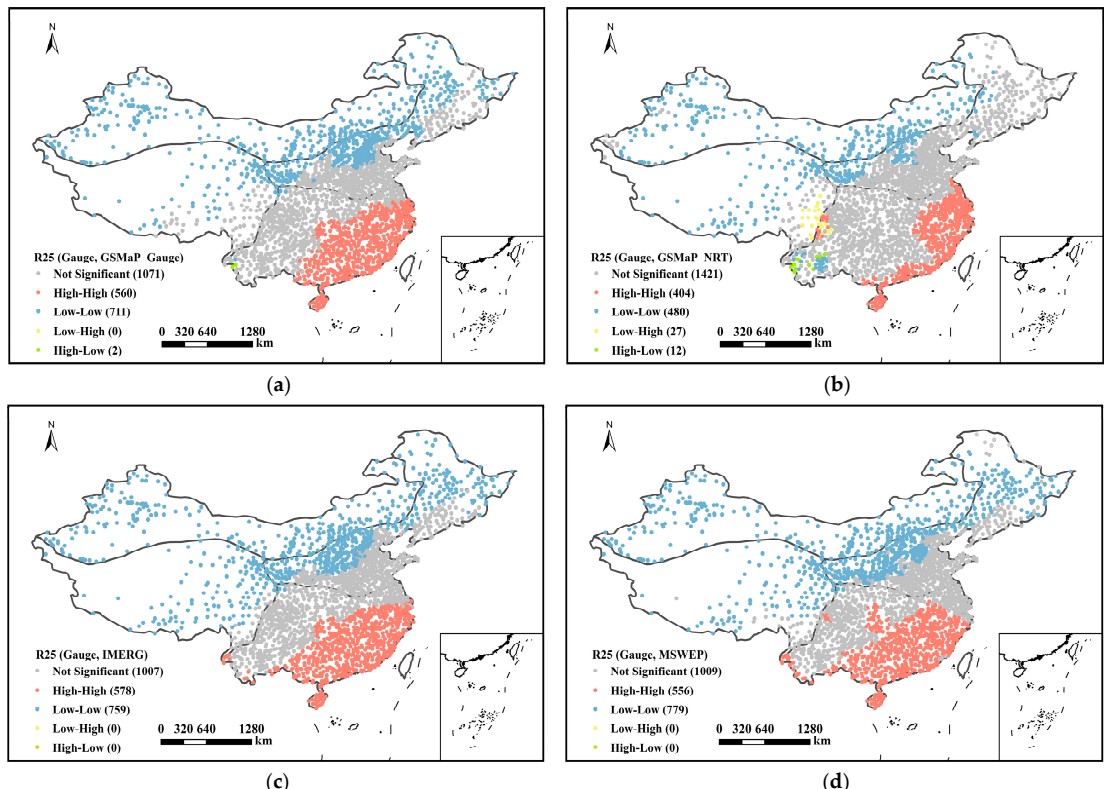

**Figure 16.** The LISA cluster maps of R25 between the gauge observations and the four SPPs and the number in the bracket represent the corresponding number of gauges: (**a**) GSMaP_Gauge, (**b**) GSMaP_NRT, (**c**) IMERG, and (**d**) MSWEP.

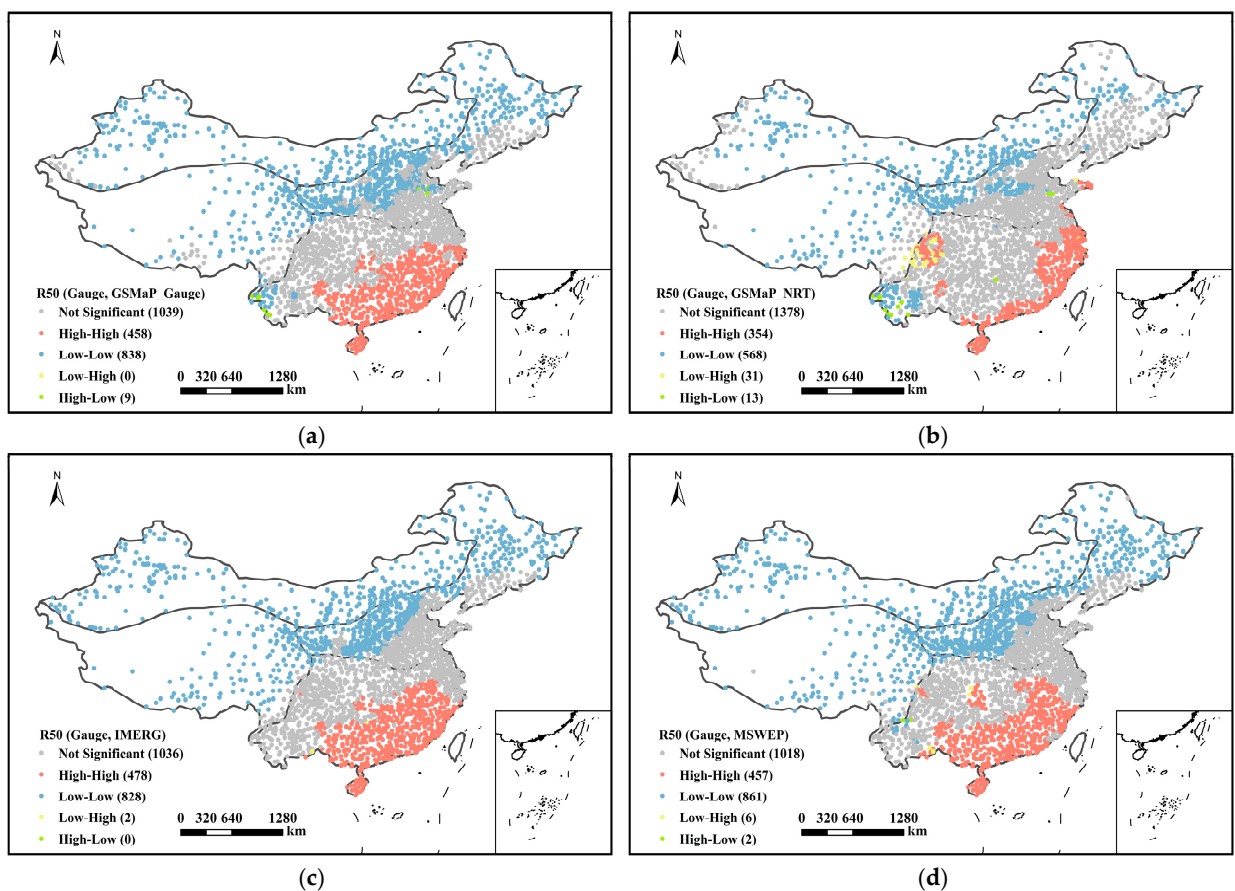

**Figure 17.** The LISA cluster maps of R50 between the gauge observations and the four SPPs and the number in the bracket represent the corresponding number of gauges: (**a**) GSMaP_Gauge, (**b**) GSMaP_NRT, (**c**) IMERG, and (**d**) MSWEP.

BMI in Four Sub-Regions

The results of the BMI of frequency indices in four sub-regions are shown in Table 6. The results demonstrate that IMERG exhibits the strongest positive spatial correlation with gauge observations across all four sub-regions, as indicated by its highest global BMI values. In contrast, GSMaP_NRT yields significantly lower global BMI values compared to the other precipitation products, particularly in NWC where the R50 global BMI values are a mere $-0.12$. Additionally, the analysis reveals weaker spatial correlation of frequency indices in NWC relative to the other regions.

**Table 6.** The global BMI of frequency indices in four sub-regions.

| Index | Sub-Region | GSMaP_Gauge | GSMaP_NRT | IMERG | MSWEP |
|-------|-----------|-------------|-----------|-------|-------|
| R25 | NC | 0.78 | 0.71 | 0.88 | 0.83 |
| | SC | 0.81 | 0.53 | 0.88 | 0.82 |
| | NWC | 0.57 | 0.28 | 0.81 | 0.51 |
| | TP | 0.52 | 0.46 | 0.68 | 0.49 |
| R50 | NC | 0.72 | 0.58 | 0.87 | 0.80 |
| | SC | 0.79 | 0.57 | 0.84 | 0.77 |
| | NWC | 0.06 | $-0.12$ | 0.68 | 0.29 |
| | TP | 0.38 | 0.30 | 0.49 | 0.26 |

## 4. Discussion

Spatial scatter plots of absolute and relative bias are commonly used to evaluate the spatial characteristics of SPPs [61,62]. These plots provide preliminary spatial correlation information. However, BMI has several advantages over spatial scatter plots.

Firstly, BMI provides a value that quantifies the spatial correlation between SPPs and gauge observations. Unlike the correlation coefficient, BMI is not site to site, but accounts for the distribution of neighbor observations by considering the special weights illustrated in Equation (5).

Secondly, the LISA cluster map not only displays the bias, but also provides information on its significance. By employing a permutation test, the significance of local BMI values can be determined, thereby indicating areas where there is a high degree of correlation or discrepancies, including underestimation or overestimation. Figure 14 shows the spatial scatter plots of the relative bias of CDD for four SPPs and gauge observations. When compared to Figure 18, the LISA cluster map (Figure 8) provides a clearer picture of exceptional sites and their special correlation relations, which is difficult to discern from the spatial scatter plots of relative bias.

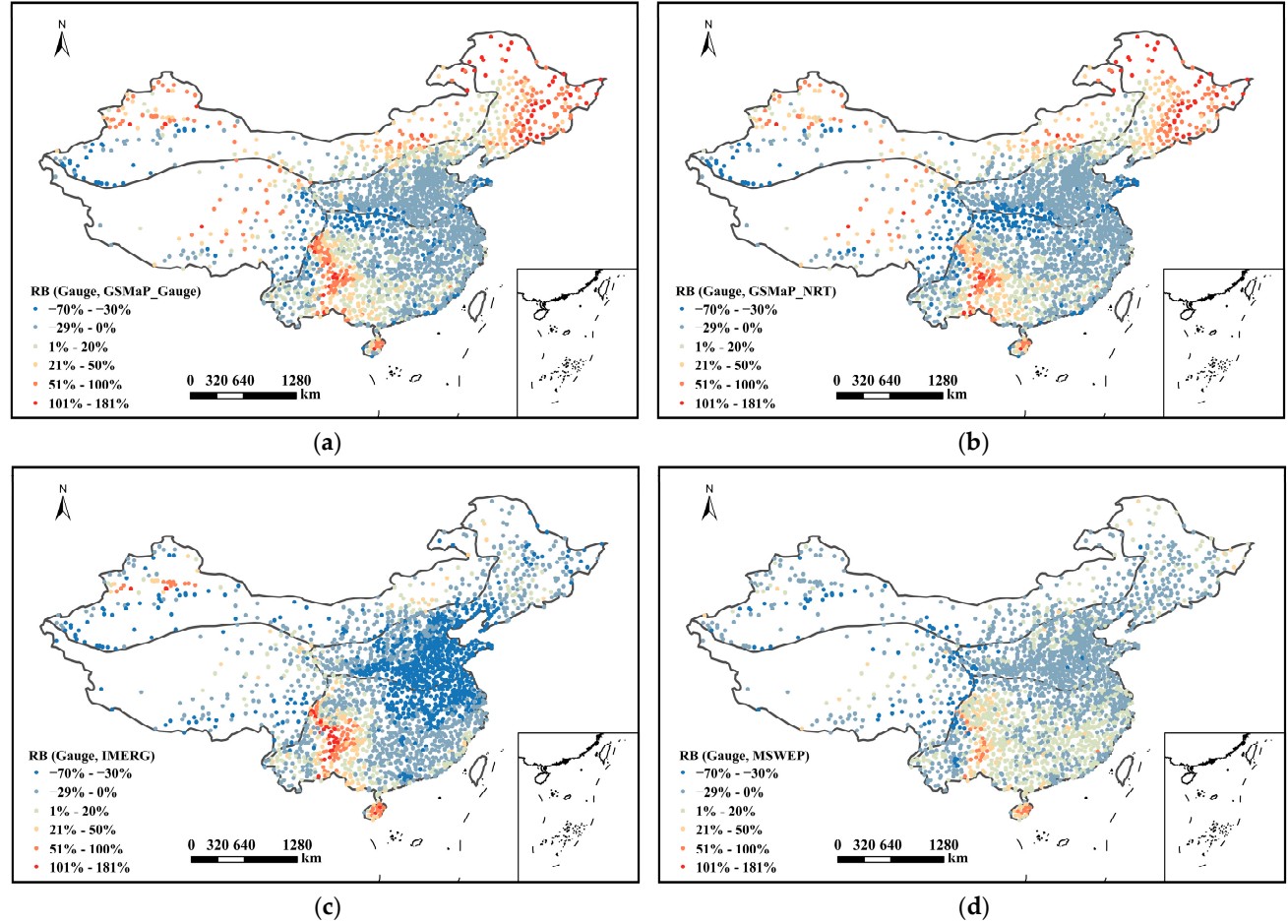

**Figure 18.** The spatial distribution scatter map of the relative bias of CDD between the gauge observations and the four SPPs: (**a**) GSMaP_Gauge, (**b**) GSMaP_NRT, (**c**) IMERG and (**d**) MSWEP.

Finally, the LISA cluster map is capable of identifying the spatial correlation relationships between SPP products and gauge observations at different scales based on their regional spatial distributions. Figure 19 illustrates the LISA cluster map of CDD between four SPP products and gauge observations in SC. Compared to Figure 14, the LISA cluster map highlights significant correlation clusters based on the regional spatial distributions of SPP products and gauge observations. However, the spatial scatter plot of absolute

bias will remain unchanged across different scales. It is important to note that BMI is similar to the correlation coefficient in that it only describes the spatial correlation between two spatial variables. However, BMI can't take into account the absolute value difference between the variables. Therefore, it is recommended to use BMI in conjunction with conventional indices that describe the absolute value difference, such as absolute deviation or relative deviation.

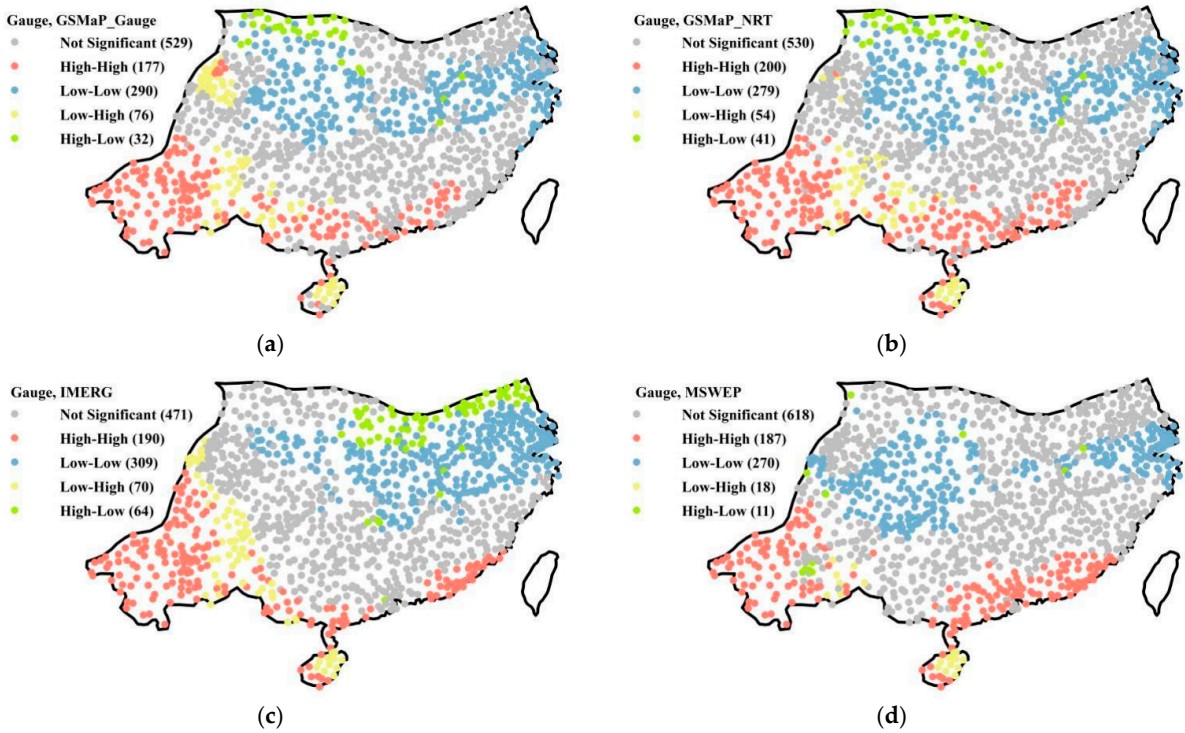

**Figure 19.** The LISA cluster maps of CDD between the gauge observations and the four SPPs in SC: (**a**) GSMaP_Gauge, (**b**) GSMaP_NRT, (**c**) IMERG and (**d**) MSWEP.

## 5. Limitations

The significance test of BMI is based on the normal distribution assumption. If the data clearly deviates from the normal distribution, it may affect the results of the significance test. Given the uneven distribution of precipitation across the Chinese mainland, the precipitation index distribution can be impacted by outliers, making it difficult to follow a normal distribution. Although some researchers also state that the test results are robust to the nonnormality of data [62], we conducted an experiment to test whether non-normality significantly affects the test results. We applied the Box-Cox method to transform the precipitation index into a normal distribution and calculated the BMI of the transformed index, and then compared the BMI of the transformed index with the original index. All precipitation indices were examined with the method, and the results for ATP are presented in Figures 20 and 21.

Compared with Figure 2 to 20, we observed that there were only slight differences in the global BMI between the Box-cox transferred precipitation indices and the original precipitation indices, which will not influence the comparative results. Compared with Figure 4 to 21, the LISA cluster map also remained unaltered. Therefore, the non-normality of the data did not affect the conclusion of our research. However, in future applications of BMI in SPPs evaluation, it will be important to examine data normality, and the influence of data non-normality should be addressed before conducting significant tests.

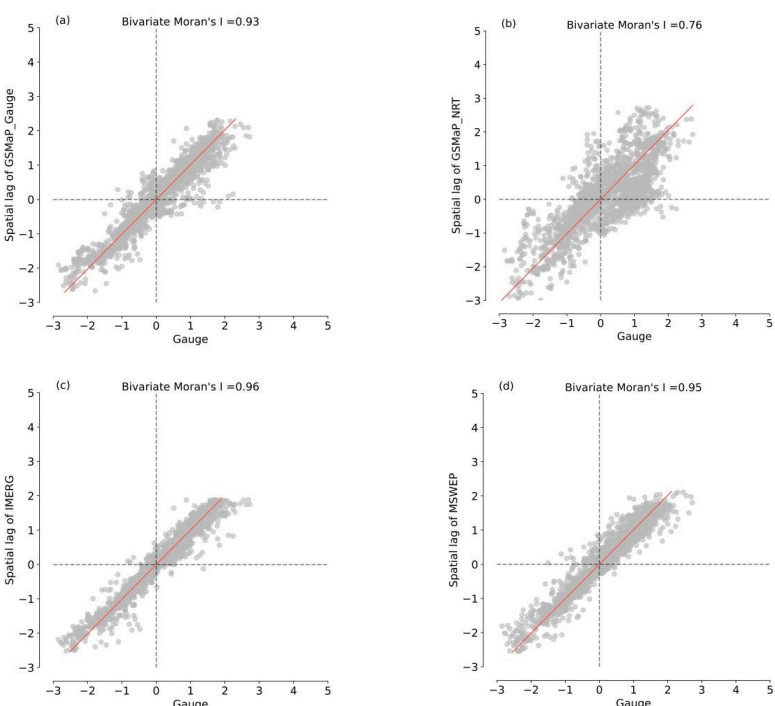

**Figure 20.** The global BMI scatter plot of Box-cox transferred ATP between the gauge observations and the four SPPs: (**a**) GSMaP_Gauge, (**b**) GSMaP_NRT, (**c**) IMERG, and (**d**) MSWEP.

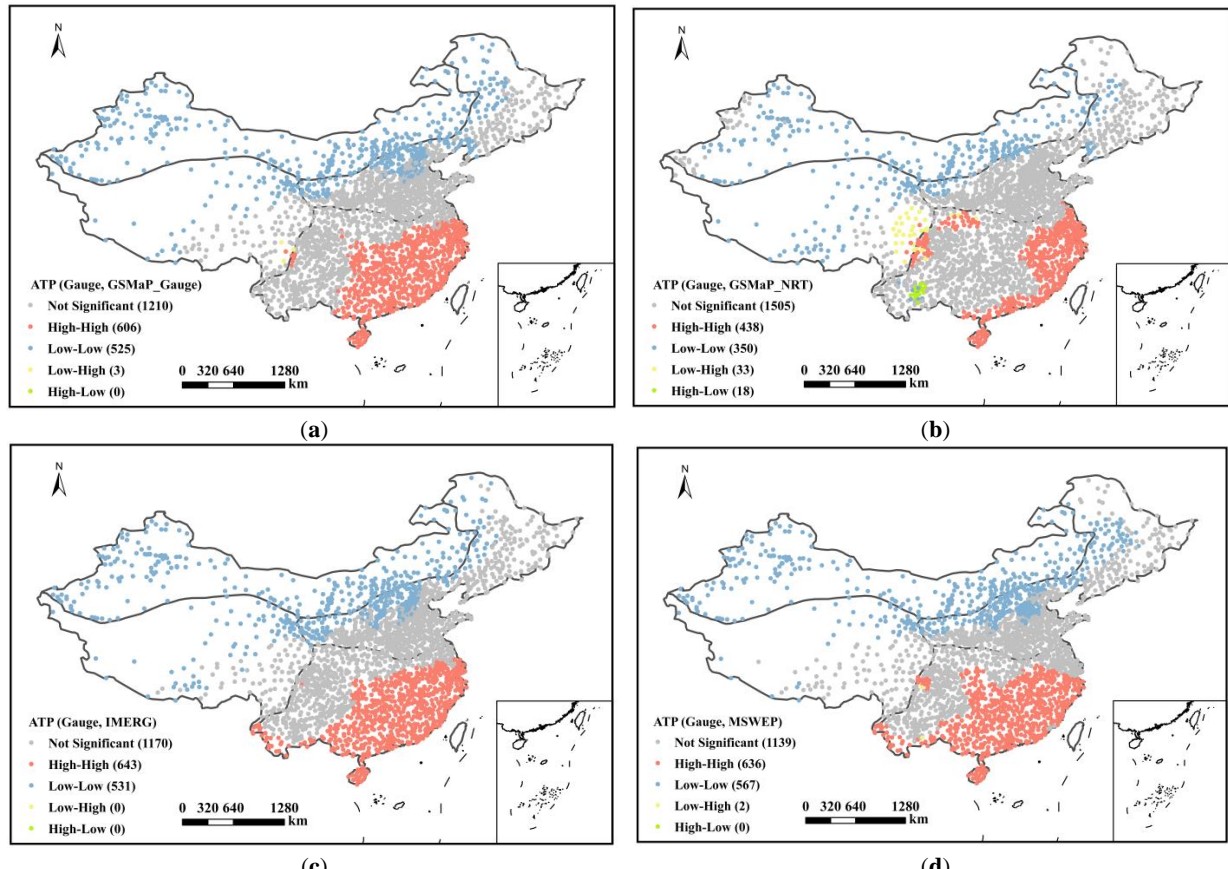

**Figure 21.** The LISA cluster maps of Box-cox transferred ATP between the gauge observations and the four SPPs and the number in the bracket represent the corresponding number of gauges: (**a**) GSMaP_Gauge, (**b**) GSMaP_NRT, (**c**) IMERG, and (**d**) MSWEP.

It should also be noted that the LISA cluster map is highly sensitive to spatial scale. As shown in Figures 8 and 19, spatial correlation clusters in SC change significantly between the two figures due to differences in spatial scale, which could lead to confusion among the public and the decision makers. Additionally, the distance threshold is a critical consideration in Bivariate Moran's I. It represents the maximum distance at which spatial relationships are expected to occur and is necessary in some circumstances to exclude non-spatially correlated factors from analysis. However, the selection of the distance threshold can significantly affect the analysis results, making it important to carefully consider an appropriate threshold. Bivariate Moran's I is also sensitive to outliers, which may lead to misjudging spatial autocorrelation when extreme values are present in the data.

## 6. Conclusions

This research aimed to evaluate the performance of four SPPs, including GSMaP_Gauge, GSMaP_NRT, IMERG V06, and MSWEP V2, over mainland China by comparing their results with daily observations from 2344 gauges. A novel evaluation method, BMI, was employed to assess the spatial correlation between precipitation indices achieved by SPPs and those obtained from gauge observations, and the results were further analyzed using LISA cluster maps. The main findings are as follows:

(1) Conventional index evaluations showed that MSWEP performed the best among the four products, with the highest correlation coefficient (0.78) and the lowest absolute deviation (1.6), relative bias ($-5\%$), and root mean square error (5). IMERG was ranked second, while GSMaP_NRT performed the worst. In terms of different sub-regions, the performance of MSWEP and IMERG also performed better, especially in the TP and NWC. Notably, IMERG showed positive deviations in all four regions, while MSWEP showed negative deviations in NC, SC, and NWC, and a positive deviation in the TP.

(2) The spatial correlation of the four SPP products with gauge observations was evaluated using BMI for total, persistent, extreme, and frequency indices. MSWEP showed the best spatial correlation relationship with the gauge observations in terms of total and persistent indices, with BMI values of 0.95, 0.89, 0.78, and 0.78, respectively. IMERG and MSWEP also showed the best spatial correlation among the extreme indices, with R95 and Rmax having BMI values of 0.84 and 0.91 for IMERG, and 0.87 and 0.88 for MSWEP, respectively. IMERG show the best performance in frequency indices, with BMI values of 0.96 and 0.92. Conversely, GSMaP_NRT had the worst spatial correlation in extreme and frequency indices.

(3) The BMI between the four SPP products and gauge observations in different regions was also calculated. The spatial correlation characteristics of SPP products differed in different regions. Generally, MSWEP showed the highest spatial correlation with gauge observations in terms of total and persistent indices in the four regions, while IMERG had the highest BMI for extreme and frequency indices. Among the four regions, the four SPPs performed high spatial correlation in NC and SC and low in TP and NWC.

In conclusion, BMI was found to be an effective tool for evaluating SPPs as it can quantitatively describe their spatial correlation with gauge observations and provide insight into the spatial trend characteristics of precipitation indices values. The LISA cluster maps were particularly useful in identifying significant overestimation or underestimation areas. These findings have the potential to advance the application of SPP products. However, the limitations of BMI should also be mentioned, such as the distribution assumption, scale effects, etc.

**Author Contributions:** Conceptualization, Y.L. and B.P.; Data curation, Y.L. and B.P.; Formal analysis, Y.L.; Software, Y.L.; Validation, Z.Z. (Ziqi Zheng), H.C. and D.P.; Writing—Original draft preparation, Y.L.; Writing—Review and Editing, B.P., Z.Z. (Zhongfan Zhu) and D.Z. All authors have read and agreed to the published version of the manuscript.

**Funding:** This research was funded by The National Natural Science Foundation of China, (51879008), The National Natural Science Foundation of China (52179003).

**Data Availability Statement:** GSMaP_Gauge and GSMaP_NRT dataset is publicly available at (http://sharaku.eorc.jaxa.jp/GSMaP/index.htm (accessed on 1 July 2021)); IMERG V06 dataset is publicly available at (https://gpm.nasa.gov/data/directory, (accessed on 1 July 2021)); MSWEP V2 dataset is publicly available at (http://www.gloh2o.org/mswep, (accessed on 1 July 2021)). The rain gauge observations data used in this study is provided by the Institute of Geographic Sciences and Natural Resources Research, Chinese Academy of Sciences and the data are not publicly available due to privacy policy.

**Acknowledgments:** The authors appreciate the valuable comments and constructive suggestions from the anonymous referees and the editors who helped improve the manuscript.

**Conflicts of Interest:** The authors declare no conflict of interest.

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
