# Peer review of "Evaluation of Four Satellite Precipitation Products over Mainland China Using Spatial Correlation Analysis"

_remotesensing, doi:10.3390/rs15071823_

Round 1

Reviewer 1 Report

Comments on “Evaluation of four satellite precipitation products over mainland China using spatial correlation analysis”

The authors evaluate GSMaP_Gauge, GSMaP_NRT, IMERG, and MSWEP against station-based data using Moran's I (BMI) that exhibits high applicability in characterizing spatial correlation and dependence. While the reviewer finds this work interesting, there are some comments to be addressed by the authors.

1.       1. Why did the authors choose MSWEP for the evaluation? MSWEP may not be a purely satellite precipitation because it combines satellite, gauges etc.

2.      2.  In terms of time scale, an evaluation of the results at the subdaily time scale would be really useful. For example, IMERG data provide subdaily precipitation.   

3.       3. Meanwhile, the authors should consider evaluating more indices of precipitation extremes in satellite products because of it is more challenging to capture precipitation extremes using satellite instruments. The authors could consider the frequency f extreme precipitation in this study.

Minor comments:

Line 117: change “And the eastern” to “The eastern”

Line 119:  change “taking Qinling Mountains Huaihe River as the boundary” to “separated by Qinling Mountains Huaihe River”

Author Response

作者使用Moran's I(BMI)根据基于站点的数据评估GSMaP_Gauge,GSMaP_NRT,IMERG和MSWEP,该数据在表征空间相关性和依赖性方面表现出很高的适用性。虽然审稿人觉得这项工作很有趣,但作者有一些评论需要解决。

答:

1. 作者为什么选择MSWEP进行评估?MSWEP可能不是纯粹的卫星降水,因为它结合了卫星,仪表等。

答:我们同意审稿人对MSWEP的评价。MSWEP综合了来自多个来源的降水估计,包括基于卫星的估计、基于仪表的观测和再分析数据。特别是,大量仪表观测的结合显着提高了其准确性。尽管如此,我们之前的研究(Li等人,2022)揭示了MSWEP的某些局限性,例如低估了极端降水。此外,这项研究还表明,IMERG在捕获极端事件方面优于MSWEP。因此,我们认为必须将MSWEP纳入我们的评估,以全面评估其能力。

参考:

李毅, 庞斌, 任敏, 等.2022种卫星降水产品捕捉北京极端降水事件性能评价[J].遥感, 14, 11(2698): <>.

2. 就时间尺度而言,在次每日时间尺度上评估结果将非常有用。例如,IMERG数据提供次日降水。

回复:我们同意审稿人的意见,即对卫星降水产品(SPPs)进行次日评估很重要,特别是对于极端降水事件。然而,实现这一目标的主要限制因素是仪表观测的可用性。作为一个拥有广大地区的发达国家,在全国范围内获得亚日尺度的仪表观测具有挑战性。因此,我们在研究中每天评估SPP。今后,我们将努力通过数据收集进行次日评估。

3. 同时,由于使用卫星仪器捕捉极端降水具有挑战性,作者应考虑在卫星产品中评估更多的极端降水指数。作者可以在这项研究中考虑极端降水的频率f。

回复:感谢您的宝贵意见。我们在研究中增加了两个频率指数,包括R25(日降水量为>25毫米)的年天数)和R50(日降水量为>50毫米的年天数)。R25 和 R50 的评估结果显示在第 3.2.4 节中。

小评论:

117行:将"和东部"改为"东部"

答:谢谢,我们已经修改了。

119路:将"以秦岭淮河为界"改为"秦岭山淮河隔开"

答:谢谢,我们已经修改了句子。

Reviewer 2 Report

The topic of the article is relevant in the field of remote sensing of precipitation. The study uses Bivariate Moran's I (BMI) and Local Indicator of Spatial Association (LISA) to provide a more comprehensive assessment of spatial patterns and relationships of four different SPPs namely GSMaP_Gauge, GSMaP_NRT, IMERG, and MSWEP with rain gauge data. Overall, the manuscript is well-written and organized. The conclusions are consistent with the evidence and arguments presented and they address the main question posed. The references are appropriate. However, the manuscript still has some deficiencies. I would recommend major revision:

               IMERG data have a temporal resolution of 30 minutes, Line 131 mentions that the resolution is 1 day, did the authors aggregate the data? Also, why did the authors use the early-run product and not the final-run product?

Lines 141 to 147 MSWEP V2 product includes merged data from rain gauges, the authors need to note that the data used evaluation is not already included in the MSWEP V2 so that their evaluation can be construed as independent.

               I agree that BMI can identify spatial patterns and relationships that may not be apparent with other conventional statistical indices. But BMI assumes that the data are normally distributed, which may not be the case for all datasets. Did the authors check the normality of the dataset?

               Also, BMI assumes that the data are stationary and that the relationship between two variables is linear. This means that if the relationship between the two variables is nonlinear or changes over time, BMI may not provide good results. Do the authors check for stationarity of the dataset? If the dataset is found to be non-stationary, appropriate statistical techniques should be used to account for non-stationarity, such as detrending or differencing the data.

               Line 153: The author uses a multi-year daily average to fill in the missing data. There are several other methods for dealing with missing rainfall such as the normal ratio method, inverse distance method etc. Why did the other choose this simple method over other methods? Provide justification with proper reference.

               Table 1: The authors should also consider other extreme rainfall indices like R20, R25, etc. which will provide additional insights into the performance of different satellite products in detecting extreme rainfall events.

Figures 4 and 5: What does the number in the bracket signify? Mention its meaning in the figure caption or in the legend.

               The font size of figures should be improved. It is not readable.

               Line 189: The authors should explain why MSWEP had the best performance over other datasets.

               Line 202-204: Similarly, please explain why the highest correlation was between SPPs and gauge observations in SC versus the NWC region.

               The authors should add limitations to this study. For instance, the Local Indicator of Spatial Association (LISA) is a local measure of spatial autocorrelation, which means that it is sensitive to changes in the spatial scale of analysis. LISA results may differ depending on the size of the study area, the distance threshold used to define spatial relationships, and other factors that affect the scale of analysis etc.

               There are several grammatical and spelling mistakes. Example: Line 66 “variety of options”

Author Response

请参阅附件。

Round 2

Reviewer 2 Report

The authors addressed the comments adequately.